# Simulation of the cement measurement based on the pulse DT neutron generator: A Monte Carlo study

Yadong Gao[1], Jiaming Li[2☉], Jichen Li[3☉], Linmao Liu[1]*

**1** College of Physics, Northeast Normal University, Changchun, Jilin Province, China, **2** Department of Medical Informatics, School of Public Health, Jilin University, Changchun, Jilin Province, China, **3** College of Electronics and Computer Engineering, Southeast University Chengxian College, Nanjing, China

☉ These authors contributed equally to this work.
* liulm888@outlook.com

**Data Availability Statement:** All relevant data are within the paper and its S1 File.

**Funding:** The author(s) received no specific funding for this work.

## Abstract

The PGNAA system for the cement measurement was simulated based on Monte Carlo method. The sizes of the moderator and reflector for the 14 MeV DT neutron generator were optimized for fast and thermal neutron outputs. The DT neutron generator was simulated at the pulse mode, and the gamma-ray detector was set as LaBr$_3$(Ce) scintillator. The characteristic peaks of the major elements (Ca, Si, Al, Fe) can be identified from the gamma-ray spectra which induced at the different time intervals of the neutron radiation. For the different thicknesses of the cement sample the ratios of the gamma-ray peaks were observed, and the result showed that when the thickness was between 20 to 30 cm, the ratios became stable. With the ratios, we can calculate the iron modulus, silica modulus and lime saturation factor.

## Introduction

Prompt Gamma Neutron Activation Analysis (PGNAA) facilities have been widely used in many industry fields, such as coal analysis, cement measurement, explosive detection, elemental analysis of special materials and etc. [1–8]. Using the facilities, the samples are irradiated by the neutrons output from the neutron sources, and the gamma rays which are induced from the thermal neutron capture reactions and fast neutron inelastic scattering reactions between the neutron sources and samples, are recorded by the gamma-ray detectors. Different energies of the gamma rays will be distributed to different channels by the electric multi-channel collectors, and then the gamma-ray spectra can be obtained. By analyzing the gamma-ray spectra researchers can identify the elements through their characteristic peaks, and the parameters which they are concerned with can be calculated [9–11].

In the PGNAA system for cement measurement the 14 MeV DT neutron generator is a potential neutron source that can be the substitution of the traditional radioisotope neutron source [12]. Comparing with the radioisotope neutron sources, the neutron generator can be turned off by the electronic control power unit, and it is more convenient for the

**Competing interests:** The authors have declared that no competing interests exist.

transportation, storage and replacement [13–21]. The problem is that the 14 MeV neutrons need to be moderated to thermal neutron energy range to induce the gamma-rays from thermal neutron capture reactions. In this study the PGNAA system based on the pulse DT neutron generator was simulated using MCNP code. By setting the DT neutron generator at pulse mode, the gamma ray spectra induced from fast neutron inelastic scattering reaction and thermal neutron capture reaction can be recorded respectively in the pulse time and the spare time. It means that the characteristic gamma-ray peaks overlapped together can be separated, and the background signal induced by the fast neutrons can also be reduced effectively. Therefore, the major elements (Ca, Si, Al, and Fe) in the cement sample can be identified more clearly in the different gamma-ray spectra. The thickness of the cement raw materials on the conveyor belt cannot be kept in a stable value. With the thickness of the cement sample increase the ratios of the peaks were calculated, and the results showed that the ratios became stable in a certain thickness range of the cement sample.

## Simulation setup

In the simulation the Monte Carlo code MCNP was used to setup the model of the experiment structure. MCNP is a general-purpose, continuous-energy, generalized-geometry, time-dependent, Monte Carlo radiation-transport code designed to track many particle types over broad ranges of energies. The commonly used editions of this code was MCNP5 and MCNPX. MCNP6 is the new edition which integrate MCNP5, MCNPX and the new CAD graphics processing code. In this paper MCNP5 code was used for our simulation. The code can be used to calculate the particle transportation of neutron, electrons and protons, and design 3-D model of the experiment structure using the input file. Using the code, the particles of the source can be defined and traced from the initialized position to the targets until the particles react with the last target. In our simulation the neutrons were emitted from the neutron source, and reacted with the structures including reflector, moderator, and measured sample. The induced gamma rays from the reactions of the inelastic scattering and thermal neutron capture were recorded in the different energy bins in the output file.

The model was composed of the DT neutron generator, the reflector, the moderator, the cement sample and the gamma-ray detector [22–27]. The DT neutron generator was developed by Institution of Radiation Technology in Northeast Normal University, and with the power unit it can be set to work at the DC or pulse mode. It was surrounded by the reflector and the moderator. The cement sample was placed on the top of the moderator. The gamma-ray detector was located above the cement sample, and was shielded by the lead and boron carbide shells to protect the detector from the neutrons and gamma rays. The schematic was shown in Fig 1.

The cross section data used in the MCNP code is from the ENDF/B-VII library, and for the moderator the scattering of the thermal neutrons was considered, so the $S(\alpha, \beta)$ card was added to the materials setup. The flux of neutrons and induced gamma rays were counted by the point detector tally (F2 card). The pulse height tally (type-8 card) was used to calculate the energy deposition of the photons in the $5'' \times 5'' \times 10''$ LaBr$_3$ scintillator crystal gamma-ray detector. For the energy spectra the Gaussian expansion was performed by the GEB function in the FT8 special tally card as follow: $FWHM = a + b\sqrt{E + cE^2}$, where E is the incident gamma-ray energy, and a, b and c are the coefficients, which were defined as a = 0.01141 MeV, b = 0.02853 MeV$^{1/2}$, and c = 0.248 MeV$^{-1}$ in this calculation [28].

For the simulation of the pulse mode the TME card was added together with the Si and Sp cards, which used to set the time intervals of the pulse. The thermal neutron flux increased with the pulse time, and at the same time the background signals caused by the fast neutrons

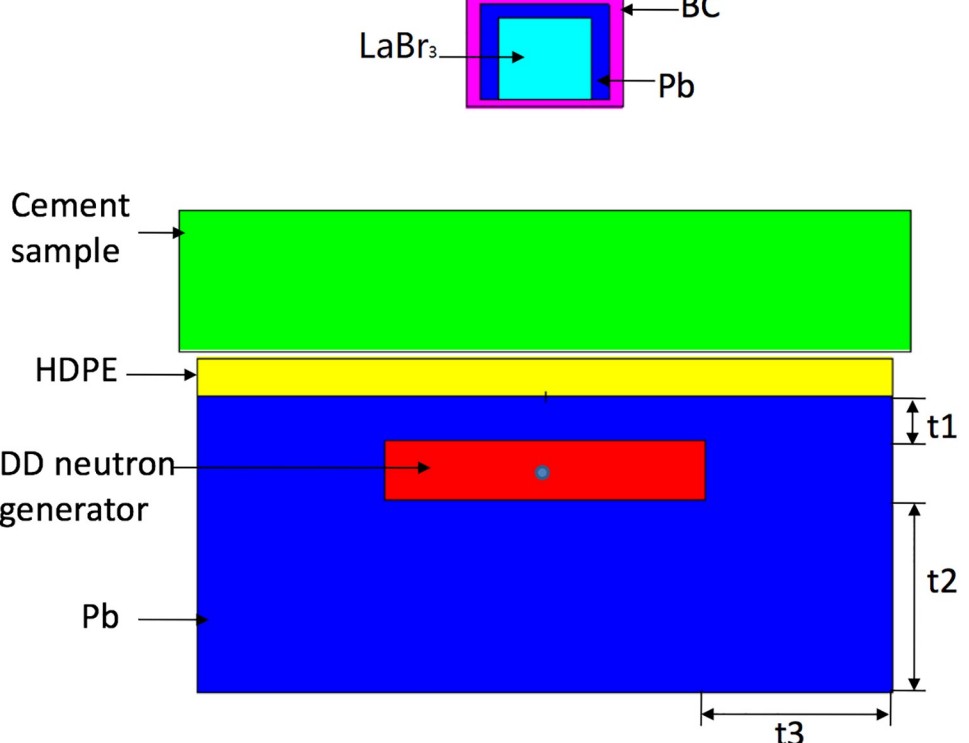

**Fig 1. A schematic of the structure for cement measurement based on DT neutron generator.**

were enhanced. In our simulation the duty circle can be set to the range of 0.2 to 0.35, and to minimize the background noise it was set to 0.2 and the pulse time was 20 μs.

The cement sample provided by the cement plant was composed of CaO, $SiO_2$, $Al_2O_3$, $Fe_2O_3$, MgO, $K_2O$, $SO_3$, $Na_2O$ and etc. The contents were measured by the X-ray fluorescence analysis spectrometer (XFA). The major compounds were CaO, $SiO_2$, $Al_2O_3$, and $Fe_2O_3$, and the contents of other compounds were very low. In the simulation the four major compounds were used, and the contents were normalized to the total mass of the four major materials as 70.5%, 21.0%, 2.5%, and 3.5% respectively.

## Reflector and moderator calculations

The calculations for the reflector and moderator were divided into two steps. In the first step the reflector was optimized to maximize the total neutron flux. The 14 MeV fast neutrons produced by the DT neutron generator were emitted in a 4π solid angle. The reflector was used to scatter the neutrons to the up direction, which can compensate the yield of the neutrons. Some materials were evaluated such as lead, bismuth, copper, nickel, graphite and etc. The thicknesses of the reflector (t1, t2, t3) were calculated, and the results were shown in Figs 2–4. It can be seen that the thickness t2 gave a big influence to the total neutron flux about 2 times increase, and when the thickness reached 20 cm the flux increased slowly. According to the simulation results, we set the thicknesses of t1, t2, t3 to 1 cm, 20 cm, and 15 cm respectively. The materials of W, WC, $W_2C$ and Cu had a better effect compared with other materials, but their weight and price were higher, so we chose lead as the material of the reflector [29–32].

In the second step the moderator was setup on the top of the reflector. The materials of the high-density polyethylene (HDPE), graphite, $H_2O$ and $D_2O$ were evaluated as the moderator

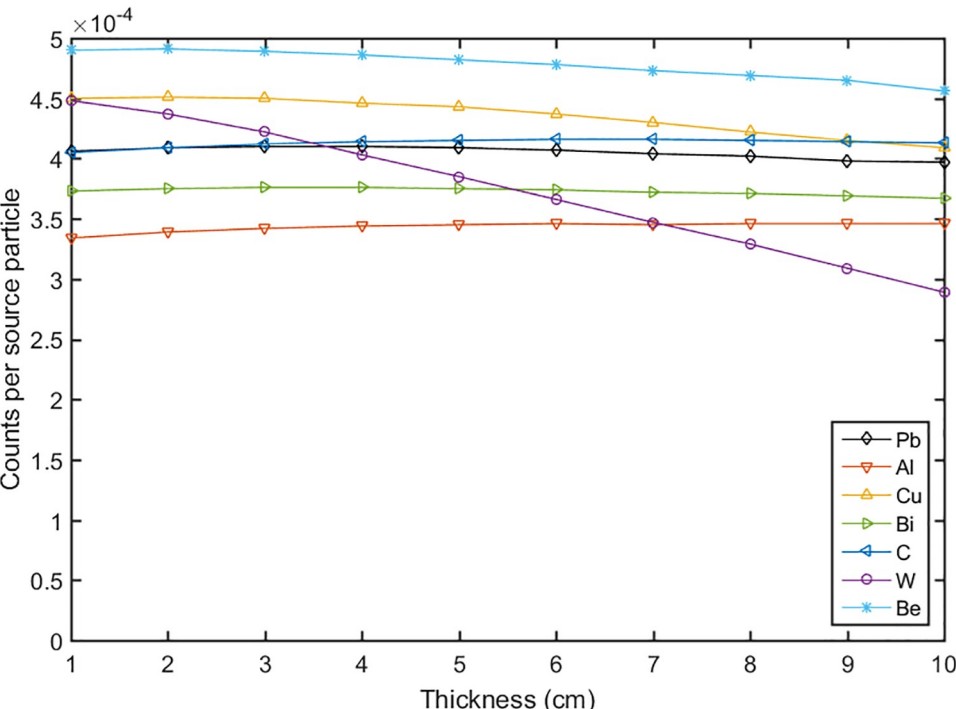

**Fig 2. The neutron flux versus the thicknesses of t1 with different materials.**

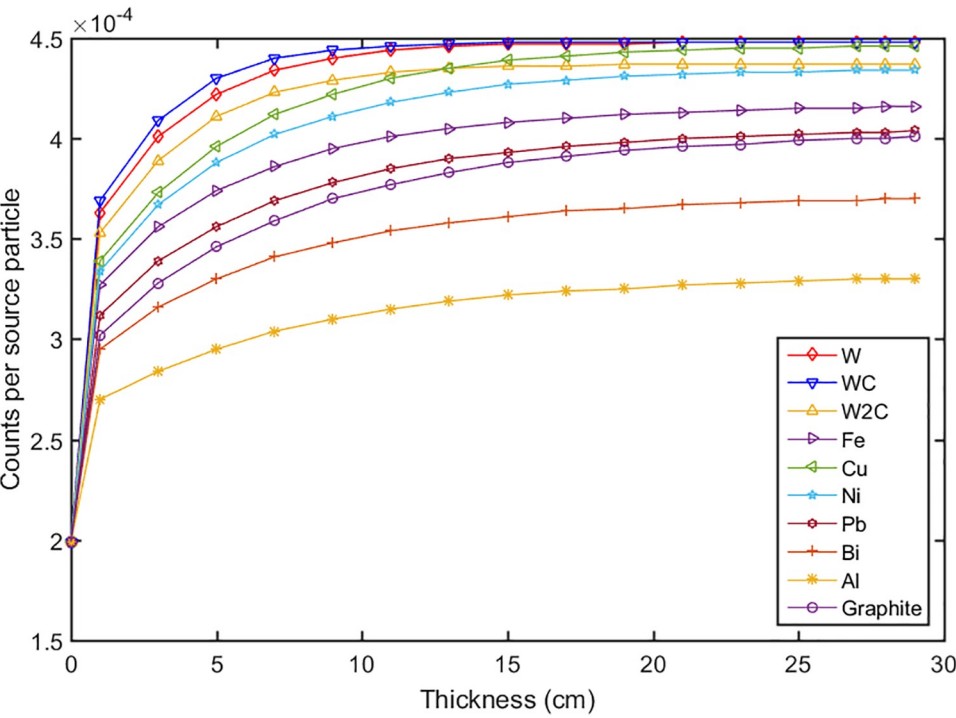

**Fig 3. The neutron flux versus the thicknesses of t2 with different materials.**

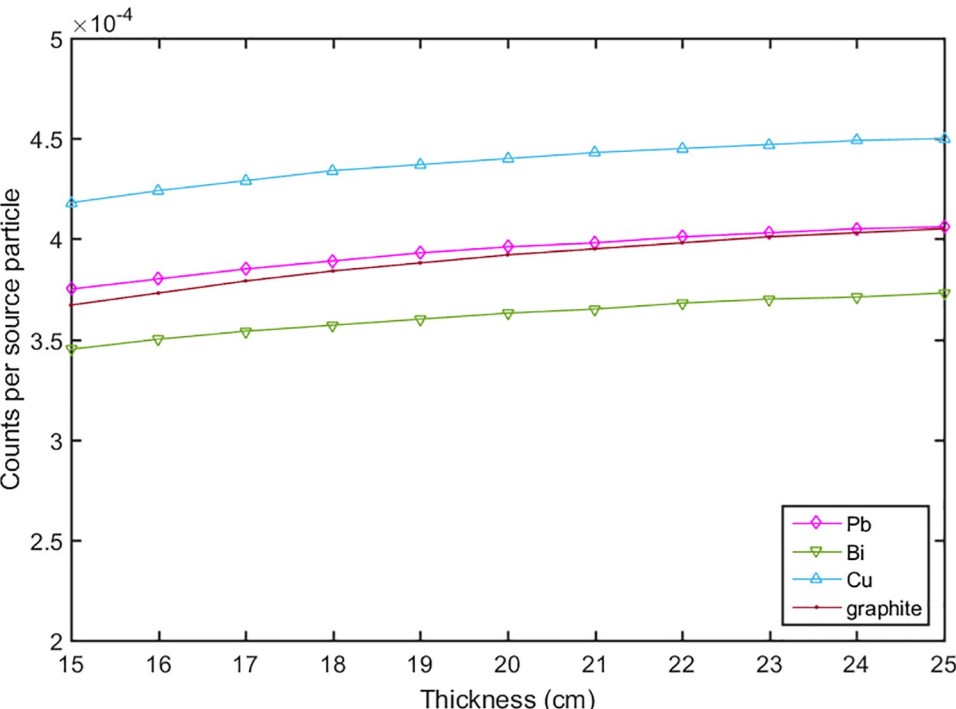

**Fig 4. The neutron flux versus the thicknesses of t3 with different materials.**

with different thicknesses to maximize the output of thermal neutrons. During the transport of the neutrons the scattering and absorption reaction occurred at the same time. At the beginning the fast and epithermal neutrons were moderated to the thermal energy level by the scattering effect, so the thermal neutrons increased. After certain thicknesses moderation, thermal neutrons decreased because the absorption effect became dominated. The results were shown in Fig 5. It can be observed that using the HDPE the flux of the thermal neutrons increased faster than other materials, and the optimal thickness was about 7 cm. The energy distributions of neutrons were compared with 1cm, 3cm, and 7cm thick HDPE in Fig 6, where it can be seen clearly that the fast neutrons were moderated to the thermal energy with the increase of the HDPE thickness.

## Gamma-ray spectra at the pulse mode

The cement samples were made up of calcium oxide (CaO), silicon dioxide ($SiO_2$), aluminum oxide ($Al_2O_3$), and iron oxide ($Fe_2O_3$) with different weight ratios. When the cement sample was irradiated by the neutrons output from the 14 MeV DT neutron generator. The gamma-ray spectra were induced by the neutron inelastic scattering (NIS) and thermal neutron capture (TNC) reactions. The characteristic peaks of the major elements (Ca, Si, Al, Fe), were related to both of the reactions, so we need both the thermal neutrons and fast neutrons.

We set the DT neutron generator at the pulse mode. For the pulse mode the duty circle was set at 0.2, which meant the time circle was divided into 20 μs fast neutron emitting time, and 80 μs spare time. At the formal time interval, the 14 MeV fast neutrons kept emitting, and at the following 80 μs the fast neutrons were moderated to thermal neutrons by the moderator and the cement sample. The gamma-ray spectra at the different time intervals were tallied as shown in Figs 7 and 8. The spectrum induced by the 14 MeV fast neutrons was called the fast

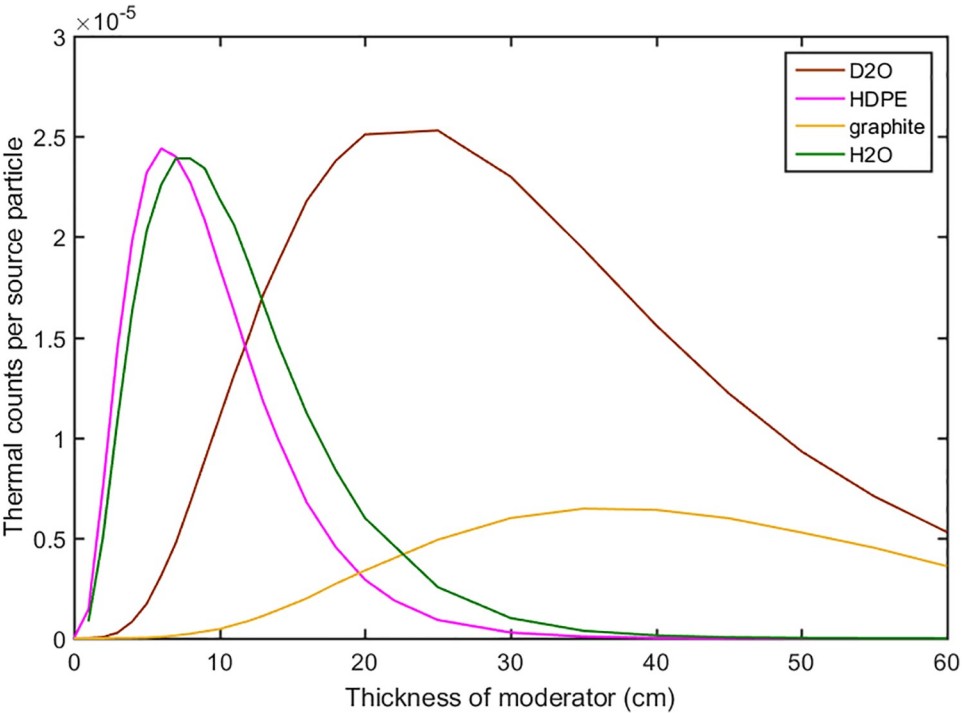

**Fig 5. The thermal neutron flux versus the thicknesses of moderator with different materials.**

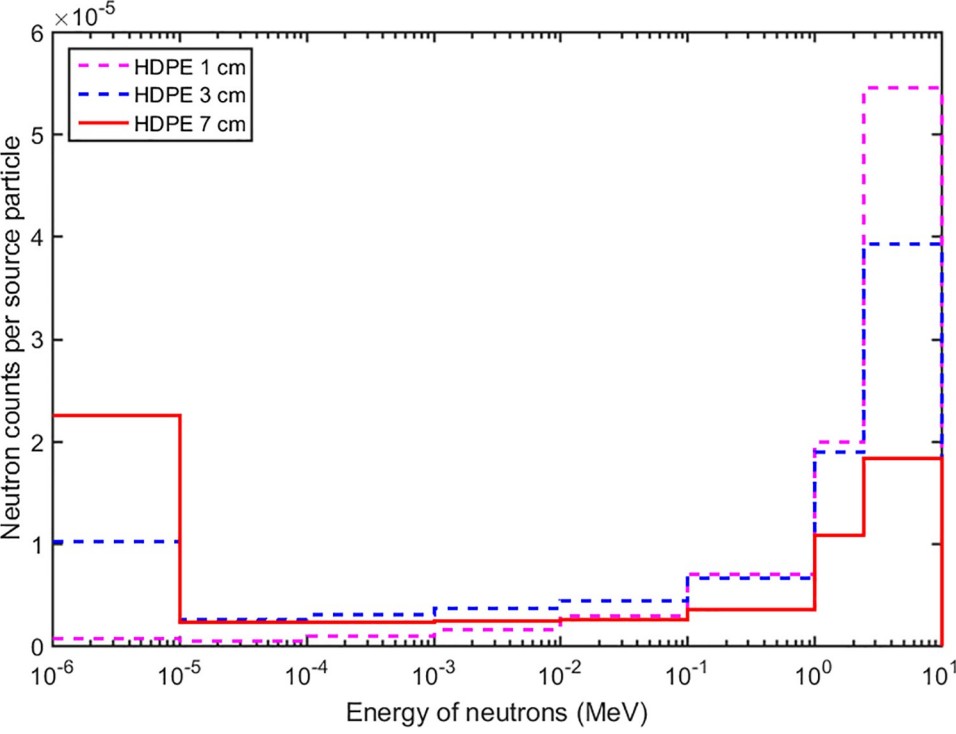

**Fig 6. The energy distributions of neutrons with different thicknesses of HDPE increase.**

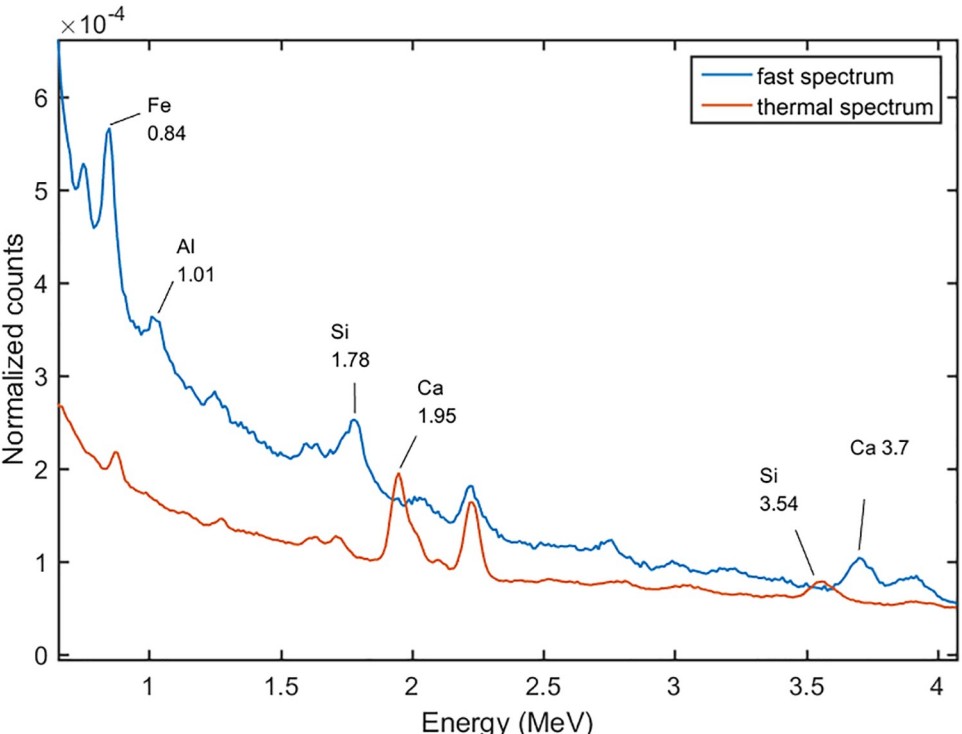

**Fig 7. The gamma-ray spectra at different time intervals at the energy range of 0 to 4 MeV.**

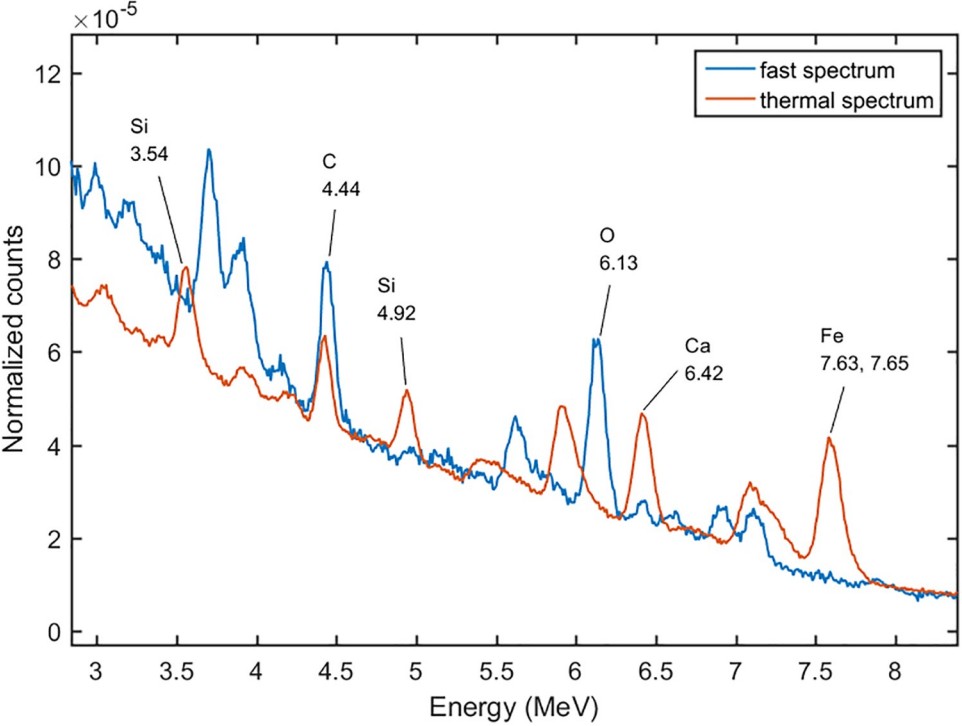

**Fig 8. The gamma-ray spectra at different time intervals at the energy range of 3 to 8 MeV.**

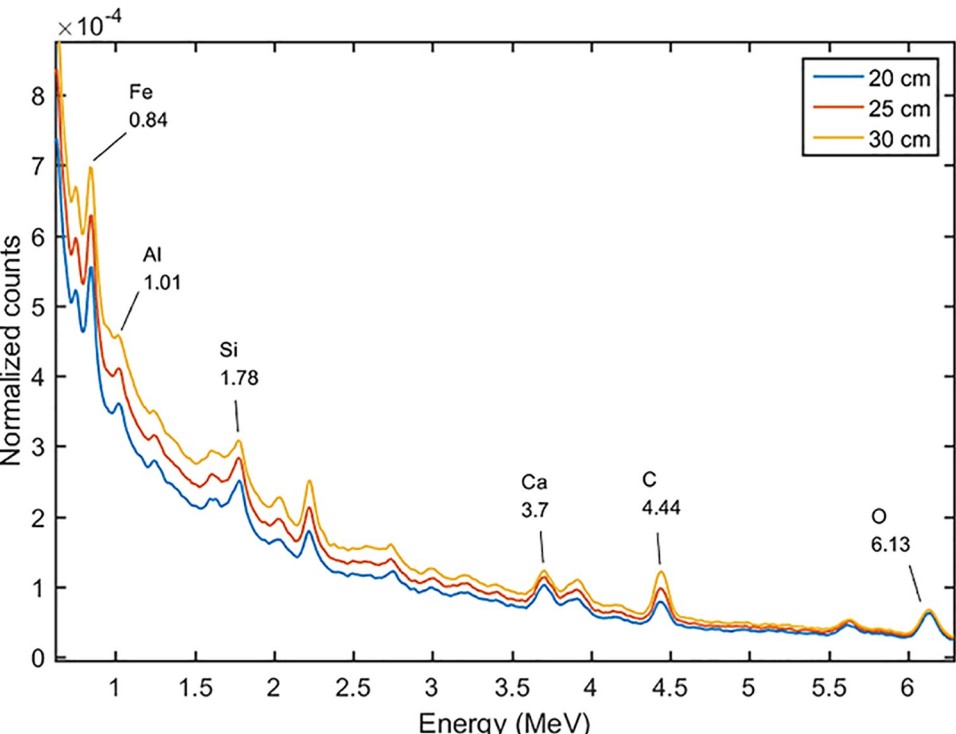

**Fig 9. The fast spectra with different thicknesses.**

spectrum, and the other one was called the thermal spectrum. It can be seen that the 0.85 MeV of Fe, 1.01 MeV of Al, 1.78 MeV peak of Si, and 3.7 MeV peak of Ca, which induced by the inelastic scattering reactions between the fast neutrons and cement sample, were shown in the fast spectrum, and the 3.55 MeV of Si, 4.92 MeV of Si, 4.42 MeV of Ca, 6.42 MeV of Ca, 7.63 and 7.65 MeV of Fe, which induced by the thermal capture reactions, were shown in the thermal spectrum. With the pulse mode used, the peaks became more remarkable than the DC mode, because they were not overlapped by each other. For example, the 3.7 MeV peak of Ca in fast spectrum can separate with the 3.54 MeV of Si in the thermal spectrum.

## Calculation results and discussion

The thickness of the cement raw materials will affect the measurement of the characteristic peaks in the gamma-ray spectra. With the transport of the neutrons, the scattering and absorption reaction occurred at the same time. The fast and epithermal neutrons were moderated to the thermal energy range by the scattering effect, so the thermal neutrons increased. After certain thickness moderation, thermal neutrons decreased because the absorption effect became dominated. As the gamma rays were induced by the NIS and TNC reactions, the absorption of the gamma rays occurred at the same time. After certain thicknesses the absorption reaction of gamma rays became dominated, So the gamma-ray spectra increased first and then decreased.

Different thicknesses of the cement sample were studied under the pulse mode simulation (Figs 9 and 10). Characteristic peaks of the major elements (Ca, Si, Al, and Fe) were shown in Fig 11. We calculated the Ca/Fe, Ca/Si, and Ca/Al ratios using their characteristic peaks in the gamma-ray spectra. When the thickness of the cement sample was at the range between 18 to 25 cm, the ratios were stable (Fig 12).

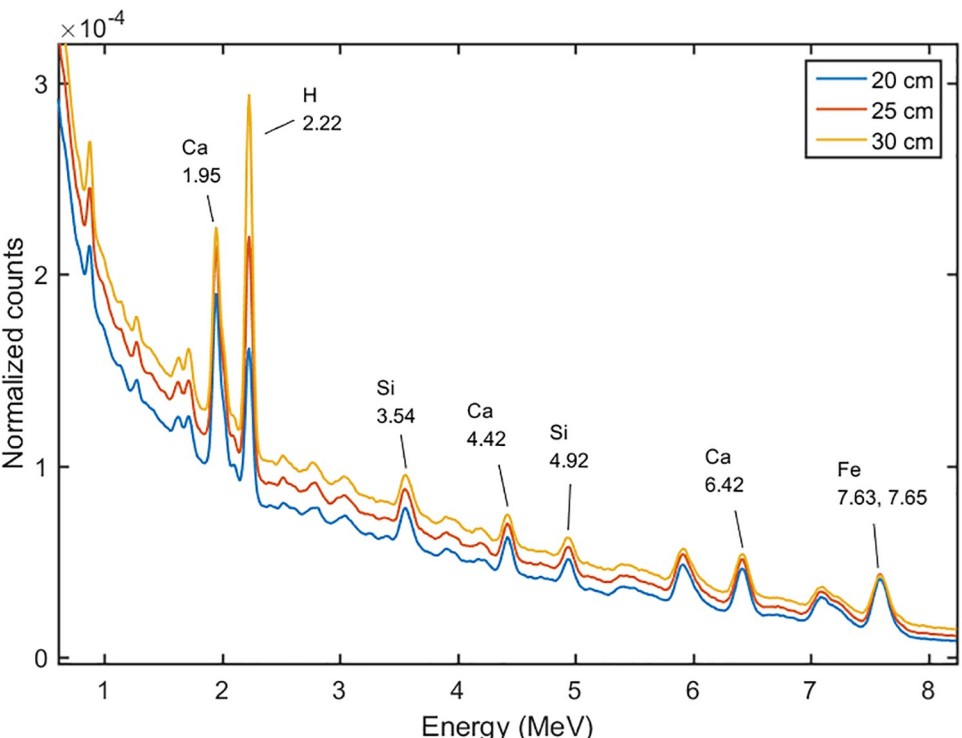

**Fig 10. The thermal spectra with different thicknesses.**

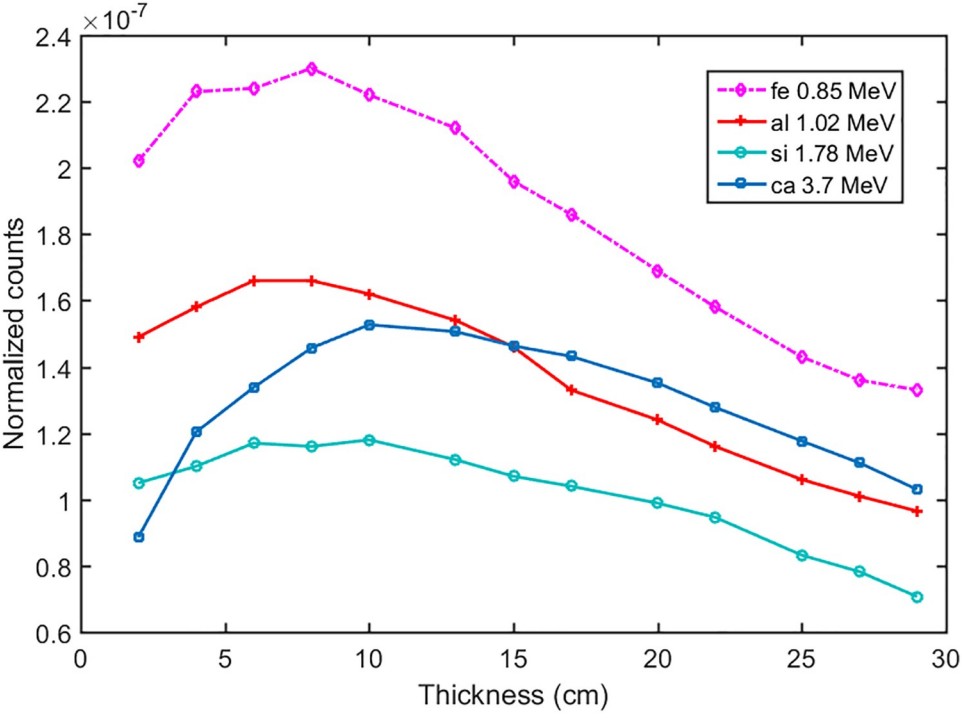

**Fig 11. The characteristic peaks as a function of the thickness of the cement samples.**

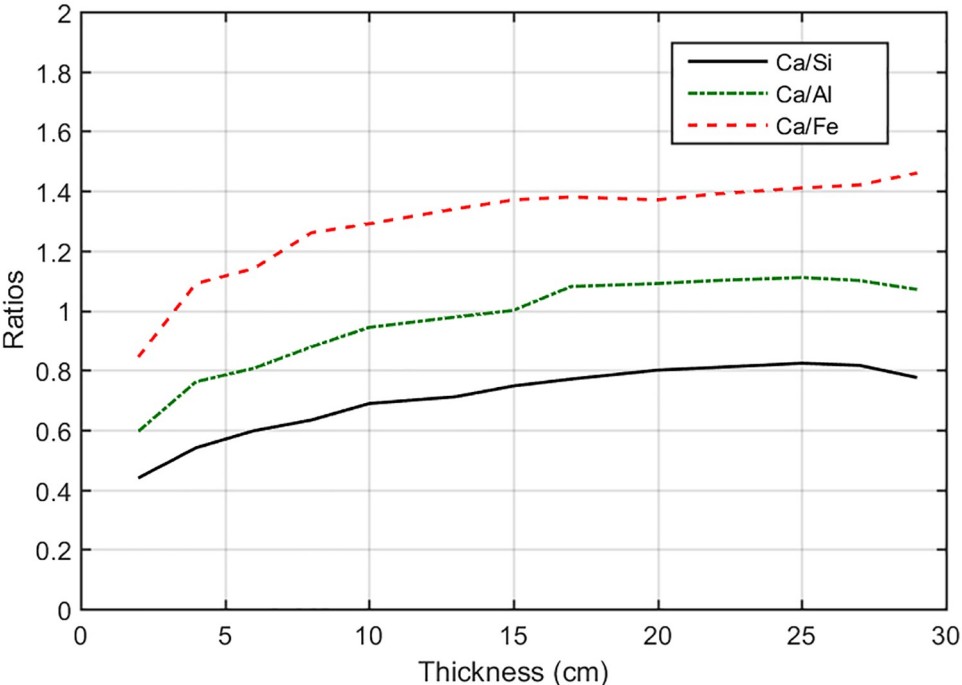

**Fig 12. The ratios of the major elements as a function of the thickness of the cement sample.**

**Table 1. Calculation of the parameters of the cement sample.**

| Thickness | KH | SM | IM |
|---|---|---|---|
| 23 cm | 0.371 | 2.473 | 1.428 |
| 25 cm | 0.369 | 2.467 | 1.431 |
| 27 cm | 0.362 | 2.462 | 1.427 |

The ratios of the elements were used to calculate the lime saturation ratio (KH), silica modulus (SM) and iron modulus (IM). The equations were listed as following, and the results were listed in Table 1, and the errors were less than 3%.

$$KH = \frac{CaO - 1.65 \times Al_2O_3 - 0.35 \times Fe_2O_3}{2.8 \times SiO_2}$$

$$SM = \frac{SiO_2}{Al_2O_3 + Fe_2O_3}$$

$$IM = \frac{Al_2O_3}{Fe_2O_3}$$

## Conclusions

By setting the DT generator at the pulse mode, the gamma-ray spectrum induced by the fast neutron inelastic scattering reaction can be separated from the spectrum induced by the thermal neutron capture reaction. For example, the Ca peak at 1.94 MeV can be separated with the

Si peak at 1.78 MeV. When the thickness of the raw materials changed, the gamma-ray peaks of the elements cannot be stable values, In the simulation we found that the ratios of the major elements (Ca, Si, Al, Fe) were stable in the range 18 cm to 25 cm, so they can be used to calculate the parameters of KH, SM and IM.

## Supporting information

**S1 File.**
(DOCX)

## Author Contributions

**Conceptualization:** Yadong Gao, Linmao Liu.

**Data curation:** Yadong Gao, Jiaming Li, Jichen Li.

**Formal analysis:** Yadong Gao.

**Investigation:** Yadong Gao.

**Methodology:** Yadong Gao.

**Project administration:** Linmao Liu.

**Resources:** Jiaming Li.

**Software:** Jichen Li.

**Writing – original draft:** Yadong Gao.

**Writing – review & editing:** Yadong Gao, Jiaming Li, Linmao Liu.

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
