## [Decision Letter · Decision Letter 0]

17 Mar 2021

PONE-D-20-37133

Simulation of the cement measurement based on the pulse DT neutron generator: A Monte Carlo study

PLOS ONE

Dear Dr. Liu,

Thank you for submitting your manuscript to PLOS ONE. After careful consideration, we feel that it has merit but does not fully meet PLOS ONE’s publication criteria as it currently stands. Therefore, we invite you to submit a revised version of the manuscript that addresses the points raised during the review process.

We look forward to receiving your revised manuscript.

Kind regards,

Hui Yao

Academic Editor

PLOS ONE

Journal Requirements:

<h1>** **</h1>

Additional Editor Comments (if provided):

Reviewers' comments:

Reviewer's Responses to Questions

**Comments to the Author**

1. Is the manuscript technically sound, and do the data support the conclusions?

Reviewer #1: Partly

Reviewer #2: Yes

2. Has the statistical analysis been performed appropriately and rigorously? 

Reviewer #1: I Don't Know

Reviewer #2: Yes

3. Have the authors made all data underlying the findings in their manuscript fully available?

Reviewer #1: Yes

Reviewer #2: Yes

4. Is the manuscript presented in an intelligible fashion and written in standard English?

Reviewer #1: Yes

Reviewer #2: Yes

5. Review Comments to the Author

Reviewer #1: This study simulation the PGNAA system with a Monte Carlo method. The following points should be clarified.

1. Please provide the full name of PGNAA

2. As stated by the author, PGNAA is widely used to measure cement composition. So what is the difference, significance and purpose of this study? This should be clarified in the introduction part.

3. The paper did not explain how Monte Carlo method was implemented in this study. For example, methodology, software or code.

4. The conclusion part is the same as the abstract and is not informative.

Reviewer #2: General comment:

This manuscript investigated the PGNAA technique for cement measurement, which represented better accuracy rather than the traditional methods. The authors measured the major elements in cement samples by the new technique and discussed the results well. There are some minor issues that should be revised by authors before possible publication.

Detailed comments:

1. Please check through the manuscript for Grammar issue, the English writing and some formats should be improved.

2. Line 70-71, please add some demonstration for choosing the duty circle to 0.2 and the pulse time to 20us? Why did you select these two parameters?

3. The cement type used in this study and the preparation process of the cement sample should be introduced.

4. The figure quality should be improved, the figure is not clear for readers.

5. Please check Figure 12, the axis names were not included.

6. For the results and discussion, why the gamma-ray spectra increased first and then decreased? Please give some demonstration.

6. PLOS authors have the option to publish the peer review history of their article (what does this mean?). If published, this will include your full peer review and any attached files.

Reviewer #1: No

Reviewer #2: No

---

## [Author Response · Author response to Decision Letter 0]

5 May 2021

Reviewer #1: This study simulation the PGNAA system with a Monte Carlo method. The following points should be clarified.

1. Please provide the full name of PGNAA

Response: The full name of PGNAA is Prompt Gamma Ray Neutron Activation Analysis. I have added this part in the revised manuscript.

2. As stated by the author, PGNAA is widely used to measure cement composition. So what is the difference, significance and purpose of this study? This should be clarified in the introduction part. 

Response: I have added the content to clarify the purpose of this study with more details in the revised manuscript as following. 

 In this study the PGNAA system based on the pulsed DT neutron generator was simulated using MCNP code. By setting the DT neutron generator at pulse mode, the gamma ray spectra induced from fast neutron inelastic scattering reaction and thermal neutron capture reaction can be recorded respectively in the pulse time and the spare time. It means that the characteristic gamma-ray peaks overlapped together can be separated, and the background signal induced by the fast neutrons can be reduced effectively. Therefore, the major elements (Ca, Si, Al, and Fe) in the cement sample can be identified more clearly in the different gamma-ray spectra. The thickness of the cement raw materials on the conveyor belt cannot be kept in a stable value. With the thickness of the cement sample increase the ratios of the peaks were calculated, and the results showed that the ratios became stable in a certain thickness range of the cement sample. 

3. The paper did not explain how Monte Carlo method was implemented in this study. For example, methodology, software or code. 

Response: Yes, it is necessary to give more explanation about the code. The following part have been added to the revised manuscript.

MCNP is a general-purpose, continuous-energy, generalized-geometry, time-dependent, Monte Carlo radiation-transport code designed to track many particle types over broad ranges of energies. The commonly used editions of this code was MCNP5 and MCNPX. MCNP6 is the new edition which integrate MCNP5, MCNPX and the new CAD graphics processing code. In this paper MCNP5 code was used for our simulation. The code can be used to calculate the particle transportation of neutron, electrons and protons, and design 3-D model for the experiment structure using the input file. Using the code, the particles of the source can be defined and traced from the initialized position to the target until the particles react with the last target. In our simulation the neutrons were emitted from the neutron source, and reacted with the structures and measured sample. The induced gamma rays from the reaction of inelastic scattering and thermal neutron capture were recorded in the different energy bins in the output file. 

4. The conclusion part is the same as the abstract and is not informative. 

Response: As Reviewer suggested, I have rewritten the conclusion part and gave more detailed expression about the simulation results in the revised manuscript, and the conclusion part is also shown as following. 

 By setting the DT generator at the pulse mode, the gamma-ray spectrum induced by the fast neutron inelastic scattering reaction can be separated from the spectrum induced by the thermal neutron capture reaction. For example, the Ca peak at 1.94 MeV can be separated with the Si peak at 1.78 MeV. When the thickness of the raw materials changed, the gamma-ray peaks of the elements cannot be stable values, In the simulation we found that the ratios of the major elements (Ca, Si, Al, Fe) were stable in the range 20 cm to 30 cm, so they can be used to calculate the parameters of KH, SM and IM. 

Reviewer #2: General comment:

This manuscript investigated the PGNAA technique for cement measurement, which represented better accuracy rather than the traditional methods. The authors measured the major elements in cement samples by the new technique and discussed the results well. There are some minor issues that should be revised by authors before possible publication.

Detailed comments:

1. Please check through the manuscript for Grammar issue, the English writing and some formats should be improved.

Response: As Reviewer suggested, we have checked the manuscript from beginning to the end several times, and the corrections were marked with red words.

2. Line 70-71, please add some demonstration for choosing the duty circle to 0.2 and the pulse time to 20 us? Why did you select these two parameters? 

Response: As Reviewer suggested we added some explanation as following. 

 The thermal neutron flux increased with the pulse time, and at the same time the background signals caused by the fast neutrons were enhanced. In our simulation the duty circle can be set to the range of 0.2 to 0.35, and in order to minimize the background noise signal it was set to 0.2 and the pulse time was 20 μs.

3. The cement type used in this study and the preparation process of the cement sample should be introduced.

Response: As Reviewer suggested, we add the information about the cement sample in the revised manuscript, and show as following. 

The cement sample provided by the cement plant was composed of CaO, SiO2, Al2O3, Fe2O3, MgO, K2O, SO3, Na2O and etc. The contents were measured by the X-ray fluorescence analysis spectrometer (XFA). The major compounds were CaO, SiO2, Al2O3, and Fe2O3, and the contents of other compounds were very low. In the simulation the four major compounds were used, and the contents were normalized to the total mass of the four major materials as 70.5%, 21.0%, 2.5%, and 3.5% respectively. 

4. The figure quality should be improved, and the figure is not clear for readers. 

Response: The figures were plotted and exported in the MATLAB program. I have adjusted the picture resolution more clearly, and uploaded the figures. The figures were shown as following. 

5. Please check Figure 12, the axis names were not included. 

Response: I am very sorry for the negligence in Figure 12. The figure was edited with the axis names added. It is also shown as following. 

6. For the results and discussion, why the gamma-ray spectra increased first and then decreased? Please give some demonstration. 

Response: As reviewer suggested, I have added the demonstration on the line 139-146 and shown as following.

 The thickness of the cement raw materials will affect the measurement of the characteristic peaks in the gamma-ray spectra. With the transport of the neutrons, the scattering and absorption reaction occurred at the same time. At the beginning the fast and epithermal neutrons were moderated to the thermal energy level by the scattering effect, so the thermal neutrons increased. After certain thickness moderation, thermal neutrons decreased because the absorption effect became dominated. With the gamma rays induced by the NIS and TNC reactions, the absorption of the gamma rays occurred at the same time. After certain thicknesses the absorption reaction of gamma rays became dominate, So the gamma-ray spectra increased first and then decreased.

---

## [Editor Report · Decision Letter 1]

10 May 2021

Simulation of the cement measurement based on the pulse DT neutron generator: A Monte Carlo study

PONE-D-20-37133R1

Dear Dr. Liu,

We’re pleased to inform you that your manuscript has been judged scientifically suitable for publication and will be formally accepted for publication once it meets all outstanding technical requirements.

Kind regards,

Hui Yao

Academic Editor

PLOS ONE
---

## [Editor Report · Acceptance letter]

24 May 2021

PONE-D-20-37133R1 

Simulation of the cement measurement based on the pulse DT neutron generator: A Monte Carlo study 

Dear Dr. Liu:

I'm pleased to inform you that your manuscript has been deemed suitable for publication in PLOS ONE. Congratulations! Your manuscript is now with our production department. 

Kind regards, 

on behalf of

Dr. Hui Yao 

Academic Editor

PLOS ONE